# Distinct Regulation of Dopamine D3 Receptor in the Basolateral Amygdala and Dentate Gyrus during the Reinstatement of Cocaine CPP Induced by Drug Priming and Social Stress

**DOI:** 10.3390/ijms22063100

**Published:** 2021-03-18

**Authors:** Rocío Guerrero-Bautista, Aurelio Franco-García, Juana M. Hidalgo, Francisco José Fernández-Gómez, Bruno Ribeiro Do Couto, M. Victoria Milanés, Cristina Núñez

**Affiliations:** 1Group of Cellular and Molecular Pharmacology, Department of Pharmacology, University of Murcia, 30120 Murcia, Spain; rocio.guerrero3@um.es (R.G.-B.); aurelio.franco@um.es (A.F.-G.); jmhc@um.es (J.M.H.); franciscojose.fernandez@um.es (F.J.F.-G.); 2Instituto Murciano de Investigación Biosanitaria (IMIB), 30120 Murcia, Spain; bruno.ribeiro@um.es; 3Department of Anatomy and Psychobiology, University of Murcia, 30100 Murcia, Spain

**Keywords:** cocaine, conditioned place preference, reinstatement, dopamine D3 receptor, social stress

## Abstract

Relapse in the seeking and intake of cocaine is one of the main challenges when treating its addiction. Among the triggering factors for the recurrence of cocaine use are the re-exposure to the drug and stressful events. Cocaine relapse engages the activity of memory-related nuclei, such as the basolateral amygdala (BLA) and the hippocampal dentate gyrus (DG), which are responsible for emotional and episodic memories. Moreover, D3 receptor (D3R) antagonists have recently arisen as a potential treatment for preventing drug relapse. Thus, we have assessed the impact of D3R blockade in the expression of some dopaminergic markers and the activity of the mTOR pathway, which is modulated by D3R, in the BLA and DG during the reinstatement of cocaine-induced conditioned place preference (CPP) evoked by drug priming and social stress. Reinstatement of cocaine CPP paralleled an increasing trend in D3R and dopamine transporter (DAT) levels in the BLA. Social stress, but not drug-induced reactivation of cocaine memories, was prevented by systemic administration of SB-277011-A (a selective D3R antagonist), which was able, however, to impede D3R and DAT up-regulation in the BLA during CPP reinstatement evoked by both stress and cocaine. Concomitant with cocaine CPP reactivation, a diminution in mTOR phosphorylation (activation) in the BLA and DG occurred, which was inhibited by D3R blockade in both nuclei before the social stress episode and only in the BLA when CPP reinstatement was provoked by a cocaine prime. Our data, while supporting a main role for D3R signalling in the BLA in the reactivation of cocaine memories evoked by social stress, indicate that different neural circuits and signalling mechanisms might mediate in the reinstatement of cocaine-seeking behaviours depending upon the triggering stimuli.

## 1. Introduction

Drug addiction is a major global health problem. Despite the current opioids crisis being the principal focus of concern worldwide, the last World Drug Report (2019, WHO) indicates that cocaine use grew between 2013 and 2017. Additionally, the number of patients seeking treatment for the first time for cocaine use disorders has augmented over the past two years in Europe (WHO, 2019). Whereas this disorder is a chronically recurrent mental disease, recovering addicts frequently fail to maintain abstinence, even after long drug-free periods [1,2] and, to date, there are no effective treatments available to diminish the risk of reinstatement of psychostimulant craving and consequent relapse [2], whose rates in humans have remained stable for the last 40 years [3]. Hence, relapse prevention is a main challenge when treating addiction [1,2,3].

To study addicts’ relapse in drug use, different models of reinstatement of drug-seeking behaviour are available, among them the reinstatement of previously extinguished conditioned place preference (CPP) and self-administration [4,5]. Drug-induced CPP is widely used to evaluate the rewarding properties of drugs of abuse. In this paradigm, the pleasurable effects of addictive drugs are associated by classical conditioning with a particular environment, which then gains and maintains secondary motivational value over a prolonged period of time [6]. This preference can be extinguished and subsequently reactivated by a drug challenge or stress, which induces the retrieval of these rewarding memories associated with the drug [6,7]. Thus, CPP but not self-administration allows one to separately examine the acquisition of the cue-environment association and the expression of the subsequent approach response [8]. A differential factor between the reinstatement paradigms and the feature of addiction that these are attempting to model is that the formers can occur in a drug-free state, while in addiction, relapse implies drug-taking after an interval of abstinence [4,5]. In humans, relapse in cocaine use can be provoked by several risk factors, among them stressful stimuli and, the most powerful of all, the re-exposure to the drug itself [3]. These same triggers are used to reinstate drug-associated behaviours in animal models [1,2,3].

Some of the molecular mechanisms of neuroplasticity underpinning substance use disorder are known to be those of learning and memory processing of natural rewards but altered by drugs of abuse [9]. Chronic exposure to these substances induces durable alterations in the circuits underlying normal learning and memory processes, which are responsible for the robust associations between drugs of abuse and the context and circumstances in which they are used [10]. The dentate gyrus (DG) is a hippocampal structure that receives dopaminergic inputs from the limbic system that are proposed to be essential for rewarding memories encoding and consolidation [11]. This area participates as well in glucocorticoids-triggered retrieval of episodic memories, which are those that associate objects or experiences with the environmental or temporal context where they occur [11,12]. Likewise, the basolateral amygdala (BLA), which provides motivational value to the environmental stimuli associated with substances of abuse, receives dopaminergic projections from the midbrain [9,13]. Therefore, alterations in dopaminergic activity in the DG and BLA have been proposed to occur during the acquisition, extinction, and reactivation of drug-associated behavioural memories [12,14,15]. 

Dopaminergic mesocorticolimbic system has been related with motivation in drug intake and relapse in drug-seeking behaviour [2,16]. Of all the dopaminergic receptors, the D3 receptor (D3R) seems to be uniquely implicated in reward, motivation, emotion, learning, and, by extension, drug seeking and relapse mechanisms, given its restricted distribution in the limbic system [17,18,19]. As a member of the D2-like receptor family, D3R is coupled to the Gα_i/o_ protein, and thus inhibits adenylyl cyclase and diminishes the intracellular levels of cAMP. However, D2-like receptors are also able to act through alternative signalling pathways, including the phosphoinositide 3-kinase (PI3K)-Akt [20]. The activation of Akt induces the phosphorylation of the mammalian (or mechanistic) target of rapamycin complex 1 (mTORC1). mTOR is an atypical Ser/Thr kinase that signals through two distinct multiprotein complexes, mTORC1 and mTOR complex 2 (mTORC2), each one with different targets and, thus, functions. Abundant findings support that mTORC1, through phosphorylation of P70 S6 kinase (P70S6K that, in turn, phosphorylates the ribosomal protein S6) and eukaryotic translation initiation factor 4E-binding protein 1 (4E-BP1), is crucial for the translation of proteins implicated in synaptic plasticity and has relevant consequences for addiction. The aberrant translation of synaptic proteins that stems from drugs of abuse-induced dysregulation of mTORC1 appears critical for the enhanced vulnerability to reinstatement of drug seeking behaviours [21,22,23].

We have previously reported that the administration of the D3R antagonist SB-277011-A blocked the reinstatement of cocaine-induced CPP elicited by social and physiological stressors but not by a drug prime [24,25]. It is known that the conditioned stimuli associated with drugs of abuse provoke intense *craving* and the activation of some brain areas, such as the BLA [17], which is critical in relapse [2,26]. The BLA interacts with the hippocampus and both contribute to the retrieval of spatial or contextual memory [27], which is processed by the DG [11]. Therefore, the goal of the present work was to study in the BLA and DG the possible alterations in the expression of some dopaminergic markers, such as D3R and the dopamine transporter (DAT), and the activity of the PI3K-Akt-mTORC1 pathway induced by the reinstatement of cocaine CPP evoked by social stress and cocaine priming, as well as the effect of D3R blockade in these parameters.

## 2. Results

It is broadly known that every drug of abuse, given its motivational value, induces CPP when administered in a sufficient dose [4,6,28]. Concordantly, the Bonferroni post hoc test uncovered that all the mice used for this study spent significantly more seconds in the cocaine-paired chamber during post-conditioning (post-C) test than during the preconditioning (pre-C) test. After the extinction sessions, there were no differences in the time spent by animals in the drug-associated compartment during the Pre-C and extinction (ext) tests, which were significantly lower than that throughout the post-C test (Figure 1B–E and Figure 2B–E).

### 2.1. D3 Antagonism Did Not Prevent the Reinstatement of the CPP Induced by a Cocaine Prime but Did Block the Reinstatement of the CPP Induced by Social Defeat

Administration of a noncontingent priming injection of the drug of abuse can reinstate drug-seeking behaviour [29,30]. We first evaluated the ability of a single dose of cocaine (12.5 mg/kg, i.p.) to induce the reinstatement of CPP. One-way ANOVA with repeated measures revealed significant differences between treatments for all the series of animals (F (2.39,31.07) = 19.33, *p* < 0.001 for vehicle (veh) + saline; F (1.49,28.36) = 17.56, *p* < 0.001 for veh + cocaine (coc) prime; F (1.74,17.41) = 17.11, *p* < 0.001 for SB 24 + coc prime; F (1.67,10.04) = 57.16, *p* < 0.001 for SB 48 + coc prime). The Bonferroni post hoc test revealed that a cocaine prime before the reinstatement (reinst) test provoked a significant increment of the time that animals passed in the drug-associated compartment when compared with the pre-C and ext tests (Figure 1C), while a saline injection did not induce any changes in the seconds spent by mice in this chamber during the reinst test (Figure 1B). In concordance with previous data of our laboratory [24,25], the administration of the D3R antagonist, SB-277011-A, was not able to block the cocaine prime-induced enhancement of the time that animals passed in the cocaine-paired compartment at none of the doses tested (24 and 48 mg/kg; Figure 1D,E). When the ratio of preference (time spent in the cocaine-associated compartment regarding the total time spent in both saline- and drug-paired chambers) was analyzed, one-way ANOVA with repeated measures revealed statistically significant differences between treatments for veh + saline (F (1.93,25.14) = 8.19, *p* < 0.01), veh + cocaine prime (F (2.09,39.70) = 16.46, *p* < 0.001), and SB 24 + cocaine prime (F (2.34,23.42) = 10.22, *p* < 0.001), whereas it failed to show differences between groups for SB 24 + cocaine prime group of animals (F (1.37,8.22) = 4.42, *p* > 0.05). The Bonferroni post hoc test revealed that a cocaine prime before the reinst test provoked a significant increment of the ratio of preference when compared with the pre-C and ext tests (Figure 1C’), while a saline injection did modify this ratio during the reinst test (Figure 1B’). The injection of SB-277011-A (48 mg/kg) did not antagonize the cocaine prime-induced increase in the ratio of preference (Figure 1E’). 

Additionally, extinguished CPP can be firmly reinstated by exposure to stressful stimuli such as social defeat [30]. One-way ANOVA with repeated measures revealed significant differences between treatments for all the sets of animals (F (1.73,25.99) = 30.81, *p* < 0.001 for veh + not defeat; F (2.24,49.25) = 40.09, *p* < 0.001 for veh + social defeat; F (1.30,16.90) = 12.22, *p* < 0.01 for SB 12 + social defeat; F (1.84,38.64) = 18.28, *p* < 0.001 for SB 24 + social defeat). The Bonferroni test showed that, as opposed to mice that underwent an agonistic social encounter (Figure 2B), animals that experienced one session of social stress increased the seconds spent in the cocaine-paired chamber throughout the reinst test in comparison with the pre-C and ext tests (Figure 2C). Coinciding with our previous studies [24], SB-277011-A, when administered at a 24 mg/kg dose, antagonized the increase in the time passed by mice in the drug-associated compartment during the reinst test after acute social stress (Figure 2E). This effect was not observed when the D3R antagonist was injected at a lower dose (12 mg/kg; Figure 2D). One-way ANOVA with repeated measures analysis of the ratio of preference revealed statistically significant differences between treatments for all the sets of mice (F (1.94,27.18) = 21.95, *p* < 0.001 for veh + not defeat; F (2.63,57.87) = 47.05, *p* < 0.001 for veh + social defeat; F (1.75,22.73) = 13.16, *p* < 0.01 for SB 12 + social defeat; F (1.81,38.09) = 33.35, *p* < 0.001 for SB 24 + social defeat).The Bonferroni post hoc test showed that one session of social stress augmented the ratio of preference in the reinst test in comparison with the pre-C and ext tests (Figure 2C’), in contrast to mice that underwent an agonistic social encounter (Figure 2B’). The D3R antagonist at a 24 mg/kg dose blocked the enhanced ratio of preference during the reinst test after acute social stress (Figure 2E’), which was not observed when SB-277011-A was administered at a lower dose (12 mg/kg; Figure 2D’).

### 2.2. The Administration of SB-277011-A Prior Drug Prime and Social Stress Altered D3R and DAT Expression in the BLA but Not in the Hippocampal DG

We next evaluated the expression of both D3R and DAT in the BLA and DG after reinstatement of cocaine CPP induced by a drug prime and social stress. 

One-way ANOVA showed significant differences between treatments in the BLA of primed mice (F (2,18) = 3.66, *p* < 0.05 for D3R; F (2,18) = 3.76, *p* < 0.05 for DAT). Although Bonferroni post hoc test did not reveal statistically significant changes, there seems to exist a trend in the enhancement of D3R and DAT levels in the BLA after the reinstatement of the CPP induced by a cocaine prime when compared to control animals administered with saline (Figure 3A,B). Moreover, SB-277011-A administration significantly decreased the levels of both proteins in this region (Figure 3A,B). In contrast, one-way ANOVA did not reveal significant differences between treatments in the DG for D3R and DAT expression after a cocaine prime (F (2,19) = 2.74, *p* > 0.05 for D3R; F (2,17) = 1.60, *p* > 0.05 for DAT). Concordantly, the post hoc test did not show changes in D3R and DAT expression in the DG of primed mice that reinstated the CPP treated with the antagonist or its vehicle when compared with controls (Figure 3C,D).

After the reinstatement of the cocaine-induced CPP provoked by an acute session of social stress, one-way ANOVA uncovered significant differences between treatments in the BLA (F (3,22) = 6.08, *p* < 0.05 for D3R; F (3,23) = 8.04, *p* < 0.001 for DAT). The Bonferroni post hoc test showed no changes in D3R and DAT expression (Figure 4A,B) regarding control animals. SB-277011-A administration decreased the expression of both proteins in this area (Figure 4A,B). The antagonist (24 mg/kg) significantly diminished DAT levels in the BLA in comparison with control mice (Figure 4B). On the contrary, one-way ANOVA showed no significant differences between treatment in the expression of D3R and DAT in the DG (F (3,25) = 2.44, *p* > 0.05 for D3R; F (3,24) = 0.69, *p* > 0.05 for DAT). In agreement, Bonferroni test did not reveal changes in D3R and DAT levels in the DG of socially stressed mice which reinstated the CPP in comparison with control animals and with socially defeated animals that received the D3R antagonist at both doses (Figure 4C,D).

### 2.3. The Reinstatement of Cocaine CPP Induced by Both a Drug Prime and Social Stress Decreased mTOR Phosphorylation Ratio in BLA and DG

We then studied the relative phosphorylation of mTOR in the BLA and DG after the reinstatement of cocaine CPP induced by drug priming and acute social defeat. 

One-way ANOVA uncovered significant differences between treatments in pmTOR phosphorylation ratio in the BLA of drug-primed animals (F (2,18) = 6.17, *p* > 0.01). Post hoc test showed a significant decrease in mTOR phosphorylation in this area after the reinstatement of cocaine CPP induced by a cocaine prime that was antagonised by the administration of 24 mg/kg of SB-277011-A (Figure 5A). Similarly, in the DG one-way ANOVA exposed significant differences between treatments in mTOR phosphorylation after a cocaine prime (F (2,19) = 9.49, *p* < 0.01). Although the post hoc Bonferroni test did not show significative changes, a decreasing trend in the pmTOR/mTOR ratio in DG was observed after the reinstatement of CPP provoked by a cocaine prime, and the blockade of D3R did not alter this ratio significantly in comparison with primed mice. However, pmTOR/mTOR in the DG of animals that received the D3R antagonist was significantly lower than control animals that received saline instead a drug prime (Figure 5B).

One-way ANOVA revealed significant differences in pmTOR/mTOR ratio in the BLA after the reinstatement of CPP induced by acute social stress (F (3,21) = 3.17, *p* < 0.05). The Bonferroni test showed significantly lower mTOR phosphorylation ratio in the BLA of socially stressed mice regarding controls and that the blockade of D3R seemed to antagonise this diminution in a dose-dependent manner (Figure 5C). Identically, the DG one-way ANOVA showed significant changes between treatments in the mTOR phosphorylation ratio of socially defeated mice (F (3,24) = 4.75, *p* < 0.01). The Bonferroni post hoc test revealed that pmTOR/mTOR also diminished statistically in the DG after the reinstatement of the CPP induced by social defeat and that this diminution was reversed by SB-277011-A administration prior the acute social stress session (Figure 5D).

### 2.4. Characterization of D3R-Expressing Neurons

Next, we studied by means of immunofluorescence the neuronal populations that expressed D3R in BLA and DG during drug- and social stress-induced reactivation of cocaine memories. Additionally, given that mTOR is regulated by several metabotropic and ionotropic receptors as well as different neurotrophic factors and can form two distinct complexes [23], we colocalized D3R with phosphorylated S6 (pS6; as a marker of mTORC1 pathway activity). 

Our study showed that in the BLA, the D3R-positive neurons were glutamatergic, although some GABAergic neurons also expressed this receptor (Figure 6 and Figure 7). Importantly, we found that D3R and pS6 colocalized in glutamatergic and GABAergic neurons in the BLA, supporting that D3R might directly control mTORC1 activity during the reinstatement of cocaine CPP induced by a drug prime and by social stress.

Similarly, the immunofluorescence images of the DG manifested that D3R is expressed in the glutamatergic and some GABAergic granule and mossy cells of the granular and polymorphic layers, respectively, after the reinstatement of cocaine CPP induced by a cocaine prime (Figure 8) and by social defeat (Figure 9). Relevantly, all the D3R-positive neurons co-expressed pS6, thus sustaining that D3R could directly regulate the mTORC1 pathway in the DG during the reactivation of cocaine CPP induced by cocaine priming and by social stress.

## 3. Discussion

In accordance with our previous studies [24,25], this work shows that the blockade of D3R inhibited the reinstatement of cocaine CPP induced by social stress but was not able to prevent the reactivation of cocaine CPP elicited by a drug-prime, pointing out that D3R modulates differently the reinstatement of drug-seeking behaviours depending upon the stimulus that triggers relapse. Although D3R antagonism differently affected the reactivation of cocaine memories, in the BLA of mice injected with SB-277011-A before receiving a drug prime or being exposed to social stress, a significant decrease in both D3R and DAT occurred in comparison with vehicle-injected animals, suggesting that a distinct neural circuits and signalling pathways would be involved in the different types of relapse. In parallel, the phosphorylation of mTOR in the BLA of mice that reinstated cocaine CPP after both social stress and a drug prime diminished, and that diminution was not observed when SB-277011-A was administered. Conversely, we did not find changes in D3R nor DAT levels in the DG of mice that were challenged with cocaine or were subjected to acute social defeat, suggesting that, in contrast to the BLA, direct D3R signalling in this hippocampal region is not critical in the reinstatement of cocaine-seeking behaviour. Nonetheless, in DG we observed reduced mTOR activation after cocaine CPP reinstatement provoked by both drug priming and social defeat. The administration of SB-277011-A blocked the reduction in the pmTOR/mTOR ratio after the reinstatement of cocaine CPP elicited by social stress, but not by cocaine prime in this region. 

Dopamine neurotransmission in the ventral tegmental area (VTA) projection areas, such as the nucleus accumbens (NAc), amygdala, hippocampus, and prefrontal cortex (PFC), is critical for reinstatement induced by drug priming, stress or cues [26]. Both D1R and D3R have been associated with drug reward and drug seeking, although D3R seems to be more relevant in the perpetuation of addiction-related behaviours [31]. In accordance, we observed that the administration of 24 mg/kg of SB-277011-A completely prevented the reinstatement of cocaine CPP induced by social defeat. Different groups have reported that D3R blockade antagonizes the reinstatement of cocaine seeking behaviours induced by different stressors [32,33]. Additionally, SB-277011-A and other D3R inhibitors prevented the reinstatement of drug seeking induced by a priming dose of several drugs of abuse [34,35,36,37]. Nevertheless, the ability of D3R antagonists to stop or mitigate the reactivation of cocaine CPP induced by drug priming seems to be dependent upon the dose of cocaine used [38,39]. Our data are in agreement with the study of Xi et al., (2006) [38], in which the D3R antagonist NGB 2904 did not inhibit the reinstatement of cocaine CPP elicited by 10 mg/kg of the drug but prevented this reactivation when induced by 2 mg/kg of cocaine and support the hypothesis of different neural circuits and signalling mechanisms mediating stress- or drug-induced relapse [2]. Given that D3R has the highest affinity for dopamine of all the dopaminergic receptors [40], our results and others might be explained by the inability of SB-277011-A to counteract the effect of the high dopamine extracellular concentration produced by the dose of cocaine used for priming in our experiment.

The BLA participates in the formation of associations between drugs and their conditioned stimuli, as well as in the behavioural responses to these previously conditioned cues [41]. This area is also essential in relapse [2,26]. Consequently, one of our objectives was to assess whether D3R expression was altered in the BLA during the reinstatement of cocaine CPP induced by social stress and cocaine prime and the effects of its blockade. Although not significantly, we observed that D3R levels tended to increase in the BLA of mice that reinstated cocaine CPP under both conditions. These data parallel the recent results of our laboratory revealing a significant enhancement of D3R expression in the NAc of mice that reinstated cocaine CPP after a drug prime or social defeat [25] and are in line also with studies that reported increased D3R in the reward system of human cocaine addicts and mice conditioned to cocaine cues [42,43]. This augment has been related to the association of cocaine effects with environmental contexts and not to the drug of abuse per se [43]. As proposed by others [44], our results might show an adaptive mechanism at a presynaptic level to counterbalance the enhanced extracellular dopamine concentration associated with drug priming [45] and social stress [46,47,48] responsible for the reactivation of cocaine-associated memories, by which the expression of D3R autoreceptors in dopaminergic terminals would augment in an attempt to diminish elevated synaptic DA levels’ dopaminergic neurotransmission. Postsinaptically, if the inhibitory character of D3R as a member of the D2-like dopamine receptor is assumed, its increased expression in the glutamatergic neurons of BLA would induce the inhibition of these cells. Although confusing, these data would agree with other reports indicating that the inhibition of BLA glutamatergic neurons is necessary for the reinstatement of nicotine-induced CPP [49]. Another possibility, given that D3R was expressed as well in GABAergic interneurons of the BLA, is that during the reinstatement of cocaine CPP its increased expression might lead to the inhibition of these cells and the subsequent activation of the glutamatergic neurons in this area. Supporting previous evidence shows that distinct neurobiological events would underlie stress- and drug-induced reinstatement [2]; when we administered SB-277011-A, the increase in D3R after the reinstatement of cocaine CPP provoked by a cocaine prime or social stress was antagonized. These data could be to some extent confusing, given that they might indicate that, as a result of D3R blockade, an increase in extracellular dopamine would occur, as previously suggested by Sokoloff et al., (2006) [50] and mediate in the inhibition of social stress-induced reinstatement of drug seeking. Nonetheless, different lines of evidence might endorse this assumption. On the one hand, stress has been reported to induce drug seeking despite low dopamine tone [51]. Clinical trials have also tested the effects on cocaine dependence of treatments that augmented dopaminergic transmission in the limbic system, although no conclusive results were obtained [52]. On the other hand, in the case that the inhibition of BLA glutamatergic neurons would be required for the reinstatement of CPP, the blockade of D3R would reverse that inhibition. As a whole, the aforementioned studies and our data may point out that the administration of SB-277011-A to social-defeated mice would reduce D3R expression in the BLA to prevent the reactivation of social stress-induced cocaine CPP and support the therapeutic use of D3R antagonists as a potential treatment to prevent stress-induced drug relapse.

DAT is a Na^+^/Cl^−^ monoamines transporter located in the plasma membrane of dopaminergic nerve terminals that regulates dopamine levels by quick re-uptake of the liberated neurotransmitter. Thus, DAT is a determinant factor in the modulation of the intensity and duration of dopamine action [26]. Furthermore, it is largely known that cocaine exerts its main effects in the mesocorticolimbic system by binding to DAT and inhibiting dopamine re-uptake in dopaminergic terminals from the VTA, thereby increasing dopamine extracellular concentration in its projection areas [53]. Our study revealed increments of DAT in the BLA of mice that reinstated cocaine CPP, although they were not significant. Evidence establishes that acute D3R activation leads to an increase in DAT activity [54]. Hence, the trend towards increased DAT after CPP reactivation might be mediated by D3R in order to diminish dopamine extracellular levels, as it has previously reported in the striatum [55]. The effect of D3R blockade impeding DAT enhancement after CPP reactivation supports that hypothesis. Several mechanisms have been proposed to be implicated in the modulation of DAT activity by D3R. Physical interactions between D3R and DAT have been demonstrated to reduce dopamine re-uptake, although prolonged exposure to D3R agonists seems to be needed [56]. Previous research has also shown that D3R controls DAT activity through different kinases, such as mitogen-activated protein kinase and PI3K [54], offering a more plausible explanation for our results. 

Here we show that the reinstatement of drug- and social defeat-induced cocaine CPP parallels augmented DAT and D3R that is found in both glutamatergic principal neurons and GABAergic interneurons of the BLA. This might be a compensatory response to the enhanced dopaminergic neurotransmission induced by drug priming [45] or acute social stress [46,47,48]. In addition, D3R activation seems to be essential for social stress-induced reinstatement of drug seeking, although the mechanisms underlying this effect are not well stablished. On this matter, present data support that D3R-mediated increased DAT expression might be involved in this process. Our findings might point out that, while in presynaptic dopaminergic terminals D3R activation would participate in the enhancement of DAT in order to reduce extraneuronal dopamine levels, postsynaptic D3R activation in the BLA might mediate in the inhibition of glutamatergic neurons, which could be critical for the reinstatement of social stress-induced cocaine seeking behaviour or of GABAergic interneurons, leading to glutamatergic neurotransmission disinhibition. It could be argued that the decrease in dopaminergic markers observed after D3R blockade was a side effect of this antagonism. However, if this decrease was a secondary effect of D3R blockade, that diminution would be observed in all the brain nuclei where D3R and the antagonist were found, and our study revealed no modifications in D3R nor DAT levels after SB-277011-A administration in the DG. Additionally, in the presence of an antagonist an upregulation of dopamine receptors [57,58] has been previously described. Nonetheless, we found D3R and DAT downregulation when the D3R antagonist was administered.

In the CPP paradigm, the environment paired with the drug acquires rewarding properties after repeated associations. Thus, the posterior preference for the drug-associated compartment could be conceptualized as a type of drug seeking behaviour that can be measured differently from drug taking [16]. The hippocampus is the main area that encodes the relationship between drugs and their contextual cues, and, particularly, the activity of DG is known to be vital for the acquisition and retrieval of drug-associated memories [12,59]. Additionally, the DG plays a critical role in cue-, stress-, or drug-induced reinstatement of psychostimulant and opioids seeking [60,61,62]. In agreement with our data, the administration of D1- and D2-like receptors antagonists in the DG has been reported to attenuate the reinstatement of morphine CPP [63,64]. However, although systemic D3R blockade prevented social stress-induced reactivation of cocaine CPP, alterations in D3R or DAT expression in the DG seem not to participate in this process. Likewise, no changes in D3R or DAT levels were found following the drug-induced reinstatement of cocaine CPP. Nonetheless, an indirect mechanism for D3R regulation of DG-mediated memory retrieval might be involved in this process via glucocorticoids release control. It is known that glucocorticoids induce the retrieval of episodic memory in the DG [11]. We previously reported that the increased corticosterone plasma concentration in mice that reinstated cocaine CPP after acute social stress positively correlated with the score of the reinstatement test and also that the enhanced corticosterone levels during reinstatement were reversed by D3R antagonism [24]. Thus, although D3R signalling in DG might be essential for social stress-induced reinstatement of drug seeking, further investigation is needed to disentangle its mechanistic. 

One of the signalling pathways that can be modulated by D3R is the PI3K-Akt-mTORC1 cascade [20]. mTORC1 has been implicated in drugs of abuse-induced neuroadaptations and in drug-related behaviours, such as locomotor sensitization and CPP. In addition, mTORC1 plays a pivotal role in the formation, retrieval, and reconsolidation of drug-associated memories by controlling transcription and protein synthesis at dendrites through S6 kinase 1 (S6K1) and subsequent ribosomal protein S6 phosphorylation [22]. Our data seem to indicate that the stimulation of D3R in BLA and DG during the reactivation of cocaine CPP induced by drug priming or social defeat provoked a decrease in mTOR activity. To our knowledge, mTOR phosphorylation in the BLA or hippocampus after cocaine administration has not been investigated. However, Sutton and Caron (2015) [65] reported increased mTORC1 activity in the NAc after acute cocaine administration that was related with enhanced locomotor activity. In addition, we observed augmented levels of pmTOR in the NAc after the reinstatement of cocaine CPP induced by both a drug prime and acute social stress, which were antagonized by D3R blockade [25]. While mTOR activity changes in the NAc, BLA, and DG of cocaine primed mice could be argued to be dependent on the drug, it must be noticed that socially stressed animals that did not receive cocaine showed similar modifications after CPP reactivation. In consequence, the alterations observed in these brain nuclei of social defeated or cocaine primed mice that reinstated cocaine CPP appear to be dependent on that reinstatement and not on the drug itself, as it has been previously suggested [43]. Our data revealed that the diminution of mTOR phosphorylation induced by the reinstatement of cocaine CPP was attenuated or prevented by D3R antagonism, and the colocalization of D3R and pS6 in glutamatergic afferent neurons of the BLA and in the DG granular cells support a D3R-dependent mTORC1 function in these areas. Conflicting findings about D3R modulation of mTOR activity have been published. Ours and other groups’ have reported D3R-dependent increased activity of the mTOR signalling pathway in the NAc of mice that reinstated cocaine CPP after a drug prime [25] or in several limbic areas after ketamine administration [66]. Oppositely, it has been published that D3R mediates autophagy in the striatum through mTORC1 inhibition [67]. Homo- and heterodimerization in vivo of all dopaminergic receptors with effects on native receptors signalling has also been described [68]. On the other hand, mTOR function is modulated, in addition to dopaminergic, by other metabotropic (metabotropic glutamate receptor 5) and ionotropic (*N*-methyl-d-aspartate-NMDA, α-amino-3-hydroxy-5-methyl-4-isoxazolepropionic acid-AMPA) receptors and different neurotrophic factors [23]. Nevertheless, our data seem to indicate that D3R is essential for the maintenance of mTOR basal activity during the reactivation of cocaine memories induced by a drug prime or social stress in DG and BLA. Controversial results have also been found regarding mTOR activity in the hippocampus, amygdala, or NAc after the administration of several drugs of abuse, such as tetrahydrocannabinol [69,70], alcohol [71], nicotine [72], or methamphetamine [73], and during drug-related behaviours. Increased mTOR activity was seen in the NAc after drug-induced reinstatement of cocaine [25] and alcohol [74] seeking and also in the amygdala (but not in the NAc or PFC) after morphine self-administration [75]. Moreover, levels of phosphorylated Akt, mTOR, and P70S6K were significantly enhanced in the hippocampal CA3, but not in the CA1, NAc, or VTA following morphine CPP [76], and reactivation of cocaine CPP induced by social stress did not modify mTOR phosphorylation in the NAc [25]. Altogether, ours and prior findings demonstrate that the regulation of mTOR pathway by drugs of abuse and drug-related behaviours is intricated and might be conditioned by the stimuli and the brain areas implicated in their response.

Balanced mTOR signalling activity is necessary for an adequate memory processing. mTOR signalling exerts this function by controlling transcription and synthesis of dendritic proteins that mediate in late phase long-term potentiation and synaptic plasticity [21]. Although more research has been performed about mTOR implication in drugs of abuse-induced neuroadaptations in the VTA and NAc, less is known about its function in the hippocampus and amygdala of addicted brains. It seems broadly accepted that increased mTOR activity occurs in parallel with enhanced synaptic activity [21]. In agreement, mTORC1 inhibition in the hippocampus have resulted in impaired spatial memory retrieval [77]. Nonetheless, we observed diminished mTOR phosphorylation ratio in the BLA and DG of mice that reactivated cocaine memories after a drug prime or acute social stress. According to the aforementioned investigations, our results would point out that drug- or stress-induced reinstatement of cocaine CPP would diminish synaptoplasticity processes in these areas. However, Al Ali et al., (2017) [78] found that inhibition of S6K1 promotes neurite outgrowth in vitro, and, concordantly, it has also been reported that mTORC1-S6K1 inhibition improved synaptic plasticity and learning in the hippocampus of Angelman syndrome mice [79]. Hence, more research is needed for a better understanding of mTOR function in cognitive and memory processes.

## 4. Materials and Methods

### 4.1. Animals

All surgical and experimental procedures were performed in accordance with the European Communities Council Directive of 22 September 2010 (2010/63/UE) and were approved by the local Committees for animal research (Comité de Ética y Experimentación Animal, CEEA; RD53/2013). Protocols were designed to minimize the number of experimental animals and to minimize their suffering. Male C57BL/6 mice (n = 169 at the beginning of the study; Charles River Laboratories, Sant Cugat del Vallès, Spain) initially aged 6–7 weeks were maintained on arrival in a room with controlled temperature (22 ± 2 °C) and humidity (50 ± 10%), with free access to water and food. Animals were adapted to a reversed 12 h light–dark cycle (lights off: 08:00–20:00 h) for 7 days before the beginning of the experiments. Mice were housed in groups of four in plastic cages (25 L × 25 W × 14.5 H cm) for 10 days (n = 139; used for CPP) or for 1 month (n = 8; used as nonaggressive opponents in the test of social interaction). Animals utilized for the CPP experiment were handled 5 min daily for 5 days before the beginning of the experiments. The mice used as aggressive opponents (n = 22) were housed individually in plastic cages (23 L × 13.5 W × 13 H cm) for a month before experiments to induce heightened aggression [80].

### 4.2. Drugs and Reagents

Cocaine HCl (Alcaliber, Madrid, Spain) was dissolved in sterile saline (NaCl 0.9%), SB-277011-A (*N*-[trans-4-[cyano-3,4-dihydro-2(1*H*)-isoquinolinyl)ethyl]cyclohexyl]-4-quinolinecarboxamide dihydrochloride; Tocris, St. Louis, MO, USA) was dissolved in deionized distilled water (vehicle), and all injections were administered intraperitoneally (i.p.) in a volume of 0.01 mL/g body weight.

### 4.3. Conditioned Place Preference Paradigm

Briefly, the conditioned place preference apparatus (CPP; Panlab, Barcelona, Spain) utilized to induce a reliable preference is based on that used by Valverde et al. (1996) [81] and consists of a box with two equally sized chambers (20 L × 18 W × 25 H cm) interconnected by a rectangular corridor (20 L × 7 W × 25 H cm). Distinctive visual and tactile cues distinguish the compartments: the motifs painted on the walls (either black dots or grey stripes), the floor colouring (black or grey), and the floor texture (smooth or rough). The sensory cues combination that produces a balanced choice are for walls and floor colouring and texture, respectively: (A) black dots, black smooth floor; (B) grey stripes, grey rough floor. Transparent walls are also used to minimize the time the animal spent in the corridor. The weight transducer technology and PPCWIN software allow the continuous detection and analysis of animal position throughout the test and the number of entries in each compartment. Additionally, the correct functioning of the equipment and technology was checked at the beginning of every experiment, and there was visual confirmation that the animals were placed where the system reported for the 100% of mice. The CPP experimental protocol consists of three distinct phases: a preconditioning phase, a conditioning phase, and a testing phase. Compartment assignment was counterbalanced across mice and unbiased in terms of initial spontaneous preference. All the behavioural procedures were performed at the same time of the day.

During the first phase (day 0), or preconditioning, mice were placed in the central corridor and given access to both compartments of the apparatus for 900 s, and the time spent in each one was recorded. Animals showing strong unconditioned aversion (<33% of session time) or preference (>67%) for any compartment were discarded (n = 3). After assigning the compartments, a Student’s *t*-test showed that there were no significant differences between the time spent in the cocaine-paired and the saline-paired compartments during the pre-C phase. In each group, half of the animals received the drug or vehicle in one compartment, and the other half received it in the other one (Figure 1A and Figure 2A). 

In the second phase (conditioning), guillotine doors blocked access from both chambers to the central corridor. Eight groups of animals were conditioned with 25 mg/kg of cocaine (135 animals, n = 7–30 per group). On days 1 and 3, animals received an injection of cocaine (25 mg/kg i.p.) immediately before being confined to the drug-paired compartment for 30 min and after an interval of 4 h to avoid residual effects of cocaine, received saline immediately before confinement in the vehicle-paired compartment for 30 min. The dose of 25 mg/kg of cocaine was selected on the basis of previous studies which demonstrated that it induces a robust CPP [82,83]. On days 2 and 4, animals received an injection of saline immediately before being confined to the vehicle-paired compartment for 30 min and after an interval of 4 h, received cocaine immediately before confinement in the drug-paired compartment for 30 min (Figure 1A and Figure 2A).

The third phase or post-conditioning (Figure 1A and Figure 2A) was conducted on day 5, exactly as in the preconditioning phase (free entry to each chamber for 900 s). One hundred and thirty-five animals underwent twice a week (in no consecutive days) extinction sessions for 7–8 weeks (depending on the group), which was conducted exactly as in the pre-C and the post-C tests (free access to each compartment for 900 s). The criterion of extinction was a lack of significant differences (Student’s *t*-test) in the time spent by each group in the drug-associated chamber during the ext test with regard to that in the Pre-C test. Hence, all the animals in each group underwent the same number of extinction sessions, independently of their individual scores. Once the criterion was achieved (Pre-C = ext), a new session was performed 48 h later to confirm extinction (Figure 1A and Figure 2A).

Two days after extinction was ratified, mice were subjected to an acute social defeat session or were administered with a cocaine prime (12.5 mg/kg i.p.) and then tested to evaluate the reinstatement of the cocaine-induced CPP (Figure 1A–D). The dose of cocaine prime was chosen as it has been previously shown that it elicits CPP reactivation in male mice that have undergone extinction of cocaine-induced CPP [84]. The reinstatement (reinst) test was similar to the pre-C, post-C, and ext tests, that is, free entry to each chamber for 900 s. All procedures for reinstatement were performed in a different room from that in which conditioning and tests were conducted, which constituted then a non-contingent place to that of the previous conditioning injections.

### 4.4. Experimental Group

#### 4.4.1. Experiment 1: Effect of D3R Antagonism on the Reinstatement of Cocaine-Induced CPP Evoked by a Cocaine Prime

In order to provoke the reinstatement of the cocaine-induced CPP, animals were injected with a cocaine prime (12.5 mg/kg) and 15 min later were subjected to the reinst test. A group of control animals were administered with a saline injection instead of cocaine to confirm that the reinstatement of the drug-induced CPP was due to the cocaine. The possible involvement of D3R in the reinstatement of the cocaine-induced CPP evoked by a cocaine prime was studied by means of the administration of a single dose of a D3R antagonist, SB-277011-A (24 or 48 mg/kg), 30 min before the cocaine prime. Control mice received a vehicle injection instead of the D3R antagonist (Figure 1A).

#### 4.4.2. Experiment 2: Effect of D3R Antagonism on the Reinstatement of Cocaine-Induced CPP Evoked by Social Stress

The effects of social defeat, which can be considered a type of social stress [85,86], on reinstatement of cocaine-induced CPP were assessed. For that, mice underwent an antagonistic encounter with an aggressive opponent (of equal age and body weight) that had been individually housed, had previous fighting experience, and had been previously screened for an elevated level of aggressive behaviour. Experimental mice exhibited avoidance/flee and defensive/submission behaviours after suffering the aggressive behaviour (threat and attack) of the opponent. Defeated mice always exhibit this extreme form of upright submissive behaviour [80]. This encounter lasted 15 min and took place in a neutral transparent plastic cage (23 L × 13.5 W × 13 H cm). The criterion used to define an animal as defeated was the assumption of a specific posture of defeat, characterized by an upright submissive position, limp forepaws, upwardly angled head, and retracted ears [87]. All defeated mice experienced similar levels of aggression because of the attack behaviours from the opponent, which were initiated immediately after seeing the experimental mouse (latency < 30 s). No behavioural sequelae were seen in controls or aggressive mice. Between 5 and 10 s after social defeat episode, the reinst test was performed. With the aim of demonstrating the lack of effects of the procedure itself, an additional group underwent an agonistic/nonaggressive social encounter with a conspecific mouse that was previously grouped [85]. Since this type of opponent never initiates attack, experimental mice do not experience the experience of defeat. Thus, this type of agonistic encounter can be view as a normal social interaction between two conspecific animals with a similarly low level of aggressive behaviour, and no aggression was observed during these encounters. This encounter lasted 15 min and took place in a neutral transparent plastic cage (23 L × 13.5 W × 13 H cm). 

To evaluate the implication of D3R in the reinstatement of the cocaine-induced CPP evoked by social stress, a single dose of SB-277011-A (12 or 24 mg/kg) was injected 30 min before the antagonistic encounter. Control mice were injected with vehicle instead of the antagonist (Figure 2A). 

### 4.5. Tissue Collection and Western Blot Analysis

Immediately after reinstatement, mice were killed by cervical dislocation, and the brains were removed and stored at −80 °C until use for Western blot. Brains were sliced on a cryostat and kept at −20 °C until each region of interest comes into the cutting plane. For DG and BLA study, two consecutive 500-μm coronal slides were made corresponding to approximately −1.46 to −2.18 mm from bregma for DG, and −0.94 to −1.58 mm from bregma for BLA, according to the atlas of Franklin and Paxinos (2008) [88]. Bilateral 1 mm^2^ punches of the DG and BLA were collected into Eppendorf tubes, according to the method of Leng et al. (2004) [89]. Four bilateral DG and BLA punches were placed in Eppendorf tubes containing 50 µL of homogenization solution with phosphate buffered saline (PBS), 10% sodium dodecyl sulphate (SDS), protease inhibitors, and a phosphatases inhibitors cocktail set. The tubes were immediately frozen in dry ice and stored at −80 °C until assaying. Samples were sonicated, vortexed, and sonicated again before centrifugation at 10,700× *g* for 10 min at 4 °C, and the supernatant was isolated and stored at −80 °C. Protein content was determined using the BCA method. Samples containing equal quantities of total proteins (10–20 µg, depending on the protein of interest) were separated by 7.5% or 10% SDS-polyacrylamide gel (SDS-PAGE) electrophoresis and transferred to polyvinylidene fluoride (PVDF) membranes (EMD Millipore Corporation, Temecula, CA, USA). Western blot analysis was performed with the following primary antibodies: mouse monoclonal anti-D3R (1:100; sc-136170, Santa Cruz Biotechnology Inc., Dallas, TX, USA), rat monoclonal anti dopamine transporter (DAT; 1:2000; MAB369, EMD Millipore Corporation, Temecula, CA, USA), rabbit monoclonal anti pmTOR (1:1000; #5536, Cell Signaling Technology, Danvers, MA, USA), and rabbit monoclonal anti mTOR (1:1000; #2983, Cell Signaling Technology). After three washings with TBST (tris buffer saline tween, 0.15%), the membranes were incubated 1 h at room temperature with the peroxidase-labelled secondary anti-mouse and anti-rabbit polyclonal antibodies #31460 and #31430, respectively; (1:10,000, Thermo Fisher Scientific, Waltham, MA, USA). After washing, immunoreactivity was detected with an enhanced chemiluminescent Western blot detection system (ECL Plus, Thermo Fisher Scientific) and visualized by an LAS 500 (GE Healthcare, UK) Imager. We used glyceraldehyde3-phosphate dehydrogenase (GAPDH) as loading control. The integrated optical density of the bands was normalized to the background values. Relative variations between bands of the experimental samples and the control samples were calculated in the same image. Antibodies were stripped from the blots by incubation with stripping buffer (glycine 25 mM and SDS 1%, pH 2) for 1 h at 37 °C. The blots were subsequently reblocked and probed with a rabbit monoclonal anti-GAPDH antibody (#2118; 1:5000, Cell Signaling Technology Inc., Danvers, MA, USA). The ratios of D3R/GAPDH, DAT/GAPDH, and pmTOR/mTOR were plotted and analysed. 

### 4.6. Immunofluorescence

Immediately following SD or cocaine priming reinstatement, another set of mice were deeply anesthetized with pentobarbital and then transcardially perfused with saline followed by 4% paraformaldehyde (PFD) in 0.1 M borate buffer (pH 9.5). Brains were removed and post-fixed in PFD containing sucrose (30%) for 3 h and then placed in phosphate buffered saline (PBS, 0,1 M, pH 7,4) containing 30% sucrose overnight. Twenty-five μm rostro-caudally coronal sections containing the DG and BLA were obtained using a freezing microtome (Leica, Nussloch, Germany), collected in cryoprotectant, and stored at −20 °C until processing. The atlas of Franklin and Paxinos (2008) [88] was used to identify different brain regions. Brain sections were rinsed in PBS, and an antigen retrieval procedure was applied by treating sections with citrate buffer (10 mM citric acid in 0.05% Tween-20, pH 6.0) at 90°C for 20 min before the blocking procedure. Non-specific Fc binding sites were blocked with 7% BSA/0.3% Triton-X-100 in PBS for 2 h at RT, and the sections were incubated for 60 h (4 °C, constant shaking) with the following primary antibodies: sheep polyclonal anti-phospo-S6 ribosomal protein (pS6; 1:250, ab65748, Abcam, Cambridge, MA, USA), mouse monoclonal anti-D3R (1:150, sc-136170; Santa Cruz Biotechnology Inc.), rabbit polyclonal anti-glutaminase 2 (GLS2; 1:1000, ab113509, Abcam), and chicken polyclonal anti-glutamate decarboxylase (GAD; 1:750, NBP1-02161 Novus Biologicals, Centennials, CO, USA). Alexa Fluor 488 donkey anti-sheep IgG (1:1000; A-11015, Invitrogen, Eugene, OR, USA), Alexa Fluor 555 donkey anti-mouse IgG (1:1000; A-31570, Invitrogen), Alexa Fluor 647 donkey anti-rabbit IgG (1:1000; A-31573, Invitrogen), and Alexa Fluor 647 goat anti-chicken IgG (1:1000; A-21449, Invitrogen) labelled secondary polyclonal antibodies were applied for 4 h. After washing, sections were incubated in 4, 6-diamino-2-phenylindole (DAPI, 1:25,000) for 1 min, and the sections were mounted in ProLong^®^ Gold antifade reagent (Invitrogen).

### 4.7. Confocal Analysis

The brain sections were examined using a Leica TCS SP8 (Leica, IL, USA) confocal microscope and LAS X Software (Leica Microsystems). Images from the DG and BLA were captured from low to high magnification (10× to 63× oil objective). Confocal images were obtained using 405-nm excitation for DAPI, 488-nm excitation for Alexa Fluor 488, 555-nm excitation for Alexa Fluor 555, and 647-nm excitation for Alexa Fluor 647. Emitted light was detected in the range of 405–490 nm for DAPI, 510–550 nm for Alexa Fluor 488, 555–640 nm for Alexa Fluor 555, and 647–775 nm for Alexa Fluor 647. Every channel was captured separately to avoid spectral cross-talking. The confocal microscope settings were established and maintained by Leica and local technicians for optimal resolution.

### 4.8. Data Collection and Statistical Analysis

Behavioural data were recorded automatically by PPCWIN software (Panlab, Barcelona, Spain). As these data were collected by computer, blinding to experimental group was not required. The data are expressed as the mean ± SEM. For the CPP experiments, the statistical analysis was performed using one-way ANOVA with repeated measures followed by multiple comparisons testing using the Bonferroni post hoc test to determine specific group differences. Western-blot analyses were analysed using one-way ANOVA followed by the Bonferroni post hoc test. Differences with a *p* < 0.05 were considered significant. Statistical analyses were performed with GraphPad Prism 6 (GraphPad Software Inc., San Diego, CA, USA).

## 5. Conclusions

Summarily, the present work supports that increased D3R activity in the BLA, but not in the DG, might be essential for the reinstatement of cocaine CPP induced by social stress. Concordantly, D3R antagonism prevented the reactivation of cocaine memories provoked by acute social defeat concurrently with a diminution of D3R and DAT levels. Together, these data endorse a potential therapeutic utility of D3R blockade to prevent stress-induced relapse of cocaine addicts in drug-seeking behaviours. 

## Figures and Tables

**Figure 1 ijms-22-03100-f001:**
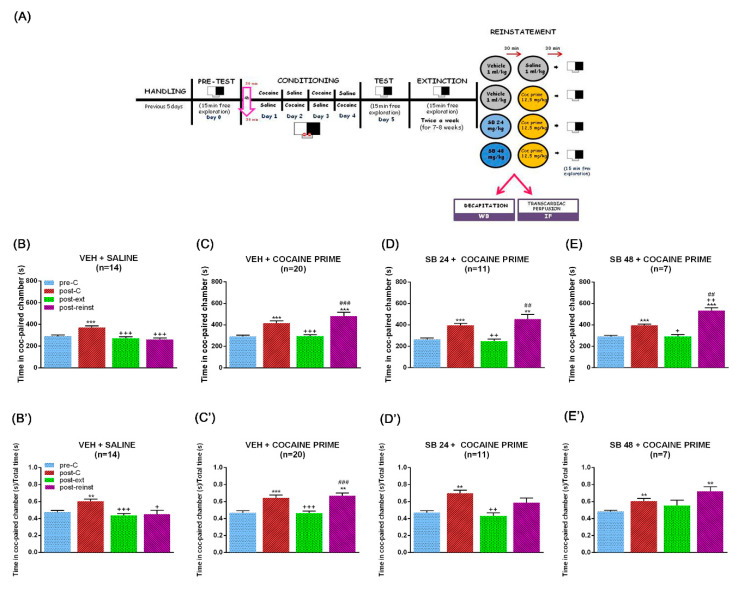
Timeline of the behavioural experimental procedure and effects of selective dopamine type 3 receptor (D3DR) blockade with SB-277011-A (24 or 48 mg/kg i.p.) on the reinstatement of conditioned place preference (CPP) induced by cocaine-priming. (**A**) Schematic representation showing the behavioural procedure. After 5 habituation and handling days, on day 0 animals were placed in the central corridor and allowed to explore the apparatus freely for 15 min. For each mouse, one chamber was randomly chosen to be paired with cocaine and the other chamber with saline. During days 1–4, animals were treated with cocaine and saline (conditioning sessions). The CPP test was conducted on day 5, exactly as in the preconditioning phase. Once achieved the criterion of extinction, a second session was performed 48 h later in order to confirm extinction. One day after the second extinction test, different groups of mice received vehicle or SB-277011-A (24 or 48 mg/kg i.p.) 30 min before saline or a cocaine prime dose. Fifteen min after saline or cocaine prime, mice were allowed to explore the apparatus freely (reinstatement test) and were sacrificed 15 min later, immediately after the reinstatement. (**B**–**E**) show the mean preference time spent in the cocaine-paired chamber during the pre-conditioning (pre-C), post-conditioning (post-C), post-extinction (post-ext) and reinstatement (post-reinst) in male mice pretreated with vehicle plus saline (**B**), vehicle plus cocaine prime (**C**) and SB-277011-A (24 or 48 mg/kg i.p.) plus cocaine prime ((**D**,**E**), respectively). (**B’**–**E’**) show the mean ratio of preference between the time spent in the cocaine-paired chamber and the total time spent in both chambers during the pre-C, post-C, post-ext, and reinst tests in male mice pretreated with vehicle plus saline (**B’**), vehicle plus cocaine prime (**C’**), and SB-277011-A (24 or 48 mg/kg i.p.) plus cocaine prime ((**D’**,**E’**), respectively).** *p* < 0.01, *** *p* < 0.001 vs. pre-C; + *p* < 0.05, ++ *p* < 0.01, +++ *p* < 0.001 vs. post-C; ## *p* < 0.01, ### *p* < 0.001 vs. post-ext. Each bar corresponds to mean ± standard error of the mean (n = 7–20 per group).

**Figure 2 ijms-22-03100-f002:**
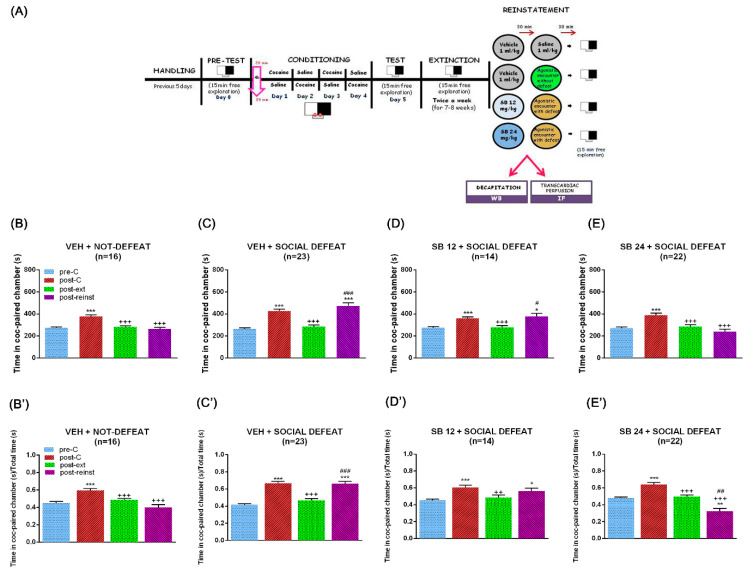
Timeline of the behavioural experimental procedure and effects of selective D3DR blockade with SB-277011-A (12 or 24 mg/kg i.p.) on the reinstatement of CPP induced by agonistic encounter with defeat. (**A**) Schematic representation showing. After 5 habituation and handling days, on day 0 animals were placed in the central corridor and allowed to explore the apparatus freely for 15 min. For each mouse, one chamber was randomly chosen to be paired with cocaine and the other chamber with saline. During days 1–4, animals were treated with cocaine and saline (conditioning sessions). The CPP test was conducted on day 5, exactly as in the preconditioning phase. Once achieved the criterion of extinction, a second session was performed 48 h later in order to confirm extinction. One day after the second extinction test, different groups of mice received vehicle or SB-277011-A (12 or 24 mg/kg i.p.) 30 min before agonistic encounter with or without defeat. Fifteen min after the agonistic encounter, mice were allowed to explore the apparatus freely (reinstatement test) and were sacrificed 15 min later, immediately after the reinstatement. (**B**–**E**) show the mean preference time spent in the cocaine-paired chamber during the pre-conditioning (pre-C), post-conditioning (post-C), post-extinction (post-ext), and reinstatement (post-reinst) in male mice pretreated with vehicle plus agonistic encounter without defeat (**B**), vehicle plus agonistic encounter with social defeat (**C**), and SB-277011-A (12 or 24 mg/kg i.p.) plus agonistic encounter with social defeat ((**D**,**E**), respectively). (**B’**–**E’**) show the mean ratio of preference between the time spent in the cocaine-paired chamber and the total time spent in both chambers during the pre-C, post-C, post-ext, and reinst tests in male mice pretreated with vehicle plus agonistic encounter without defeat (**B’**), vehicle plus agonistic encounter with social defeat (**C’**), and SB-277011-A (12 or 24 mg/kg i.p.) plus agonistic encounter with social defeat ((**D’**,**E’**), respectively). * *p* < 0.05, ** *p* < 0.01, *** *p* < 0.001 vs. pre-C; ++ *p* < 0.01, +++ *p* < 0.001 vs. post-C; # *p* < 0.05, ## *p* < 0.01, ### *p* < 0.001 vs. post-ext. Each bar corresponds to mean ± standard error of the mean (n = 14–23 per group).

**Figure 3 ijms-22-03100-f003:**
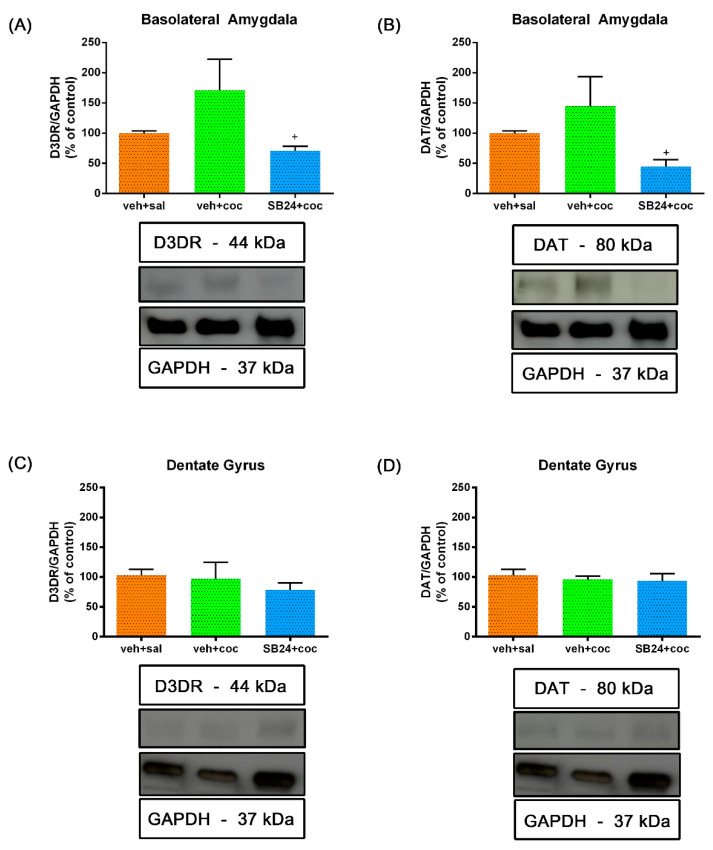
Semiquantitative analysis and representative immunoblot of dopamine type 3 receptor (D3DR) and dopamine transporter (DAT) in the basolateral amygdala ((**A**,**B**), respectively) and dentate gyrus ((**C**,**D**), respectively). Mice were conditioned to cocaine and after 60 days of abstinence they were administered with saline (veh+sal; controls), with a dose of cocaine prime (veh+coc) or with an injection of the D3DR antagonist SB-277011-A (24 mg/kg i.p.) prior to the cocaine priming (SB24+coc). A preference test was conducted giving the animals free access to both compartments of the CPP apparatus for 15 min. Mice were sacrificed immediately after the reinst test. + *p* < 0.05 vs. veh+coc. No significant changes in D3DR or DAT levels were found between groups in the dentate gyrus. Each bar corresponds to mean ± standard error of the mean (n = 5–8 per group).

**Figure 4 ijms-22-03100-f004:**
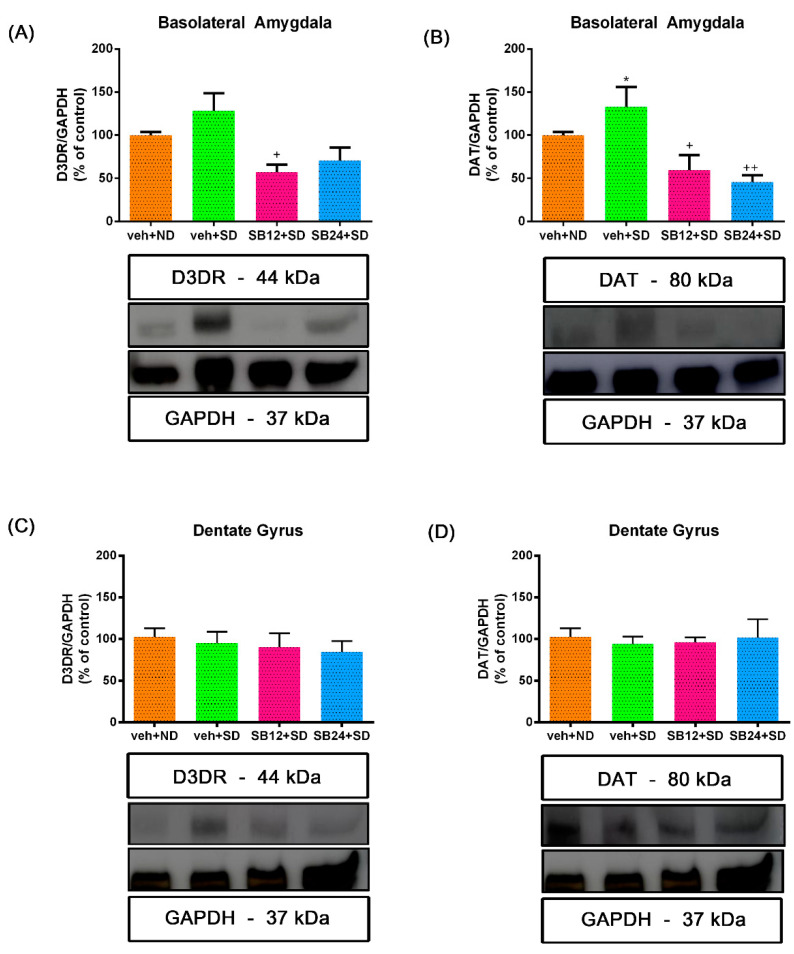
Semiquantitative analysis and representative immunoblot of dopamine receptor (D3DR) and dopamine transporter (DAT) in the basolateral amygdala ((**A**,**B**), respectively) and dentate gyrus ((**C**,**D**), respectively). Mice were conditioned to cocaine and after 60 days of abstinence they were injected with vehicle and exposed to a non-defeated agonistic encounter (veh+ND; controls) or exposed to an agonistic encounter with social defeat (veh+SD). Additionally, a group of animals was injected with the D3DR antagonist SB-277011-A (12 or 24 mg/kg i.p.) prior to the social-defeated agonistic encounter (SB12+SD and SB24+SD, respectively). A preference test was conducted giving the animals free access to both compartments of the CPP apparatus for 15 min. Mice were sacrificed immediately after the reinst test. * *p* < 0.05 vs. veh+ND; + *p* < 0.05, ++ *p* < 0.01 vs. veh+SD. No significant changes in D3DR or DAT levels were found between groups in the dentate gyrus. Each bar corresponds to mean ± standard error of the mean (n = 6–8 per group).

**Figure 5 ijms-22-03100-f005:**
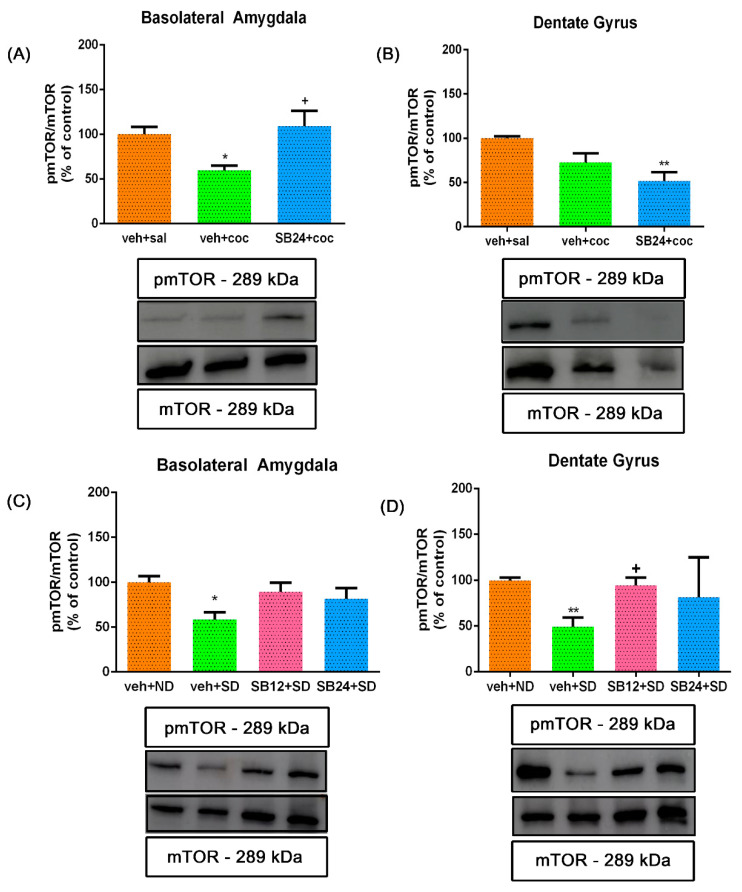
Semiquantitative analysis and representative immunoblot of phosphorylated (activated) mTOR (pmTOR) protein expression in the basolateral amygdala (**A**,**C**) and dentate gyrus (**B**,**D**). All the animals were conditioned to cocaine preference and underwent through abstinence for 60 days. (**A**,**B**) show the cohort of animals that were administered with saline (veh+sal; controls), with a dose of cocaine prime (veh+coc) or with an injection of the D3DR antagonist SB-277011-A (24 mg/kg i.p.) prior to the cocaine-priming (SB24+coc). * *p* < 0.05, ** *p* < 0.01 vs. veh+sal. (**C**,**D**) represent the set of animals that were injected with vehicle and exposed to a non-defeated agonistic encounter (veh+ND; controls) or exposed to an agonistic encounter with social defeat (veh+SD). Additionally, a group of animals was injected with the D3DR antagonist SB-277011-A (12 or 24 mg/kg i.p.) prior to the social-defeated agonistic encounter (SB12+SD and SB24+SD, respectively). A preference test was conducted by giving free access to both compartments of the CPP apparatus for 15 min to every group of animals. * *p* < 0.05, ** *p* < 0.01 vs. veh + ND; + *p* < 0.05 vs. veh+SD. Each bar corresponds to mean ± standard error of the mean (n = 5–8 per group).

**Figure 6 ijms-22-03100-f006:**
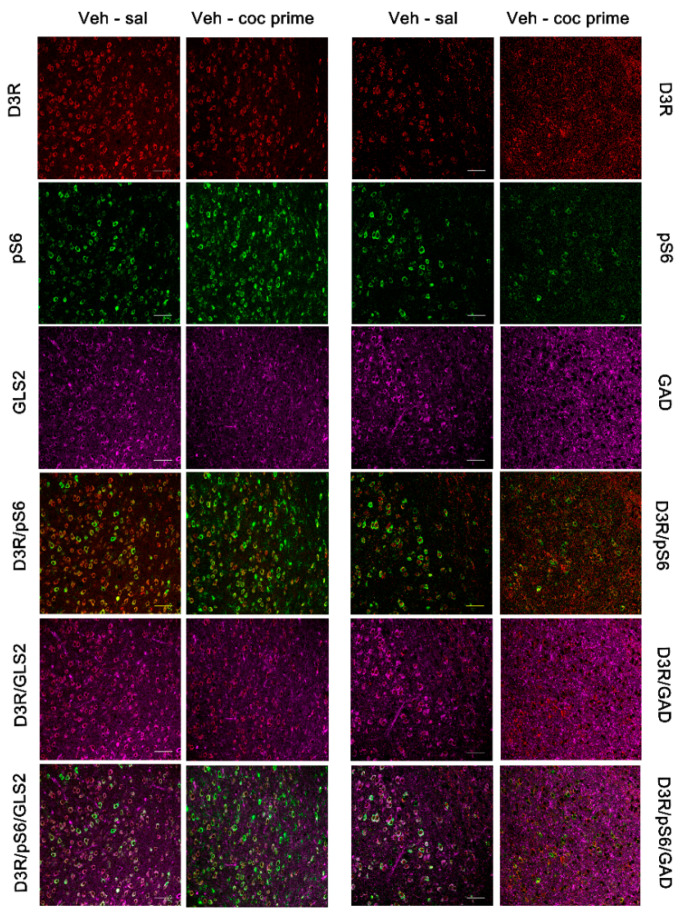
Representative confocal images showing the basolateral amygdala coronal sections immunostained for dopamine type 3 receptor (D3DR, red), phosphorylated S6 protein (activated; green), GLS2 (magenta, glutamatergic neurons), and GAD (magenta, GABAergic neurons) from controls (veh-sal) and from mice receiving a cocaine-prime dose after 60 days of abstinence (veh-coc prime). Colocalization of D3DR/GLS2, pS6/GLS2, DRD3/pS6/GLS2, D3DR/GAD, pS6/GAD, and DRD3/pS6/GAD are also shown in the figure. Scale bars: 50 μm.

**Figure 7 ijms-22-03100-f007:**
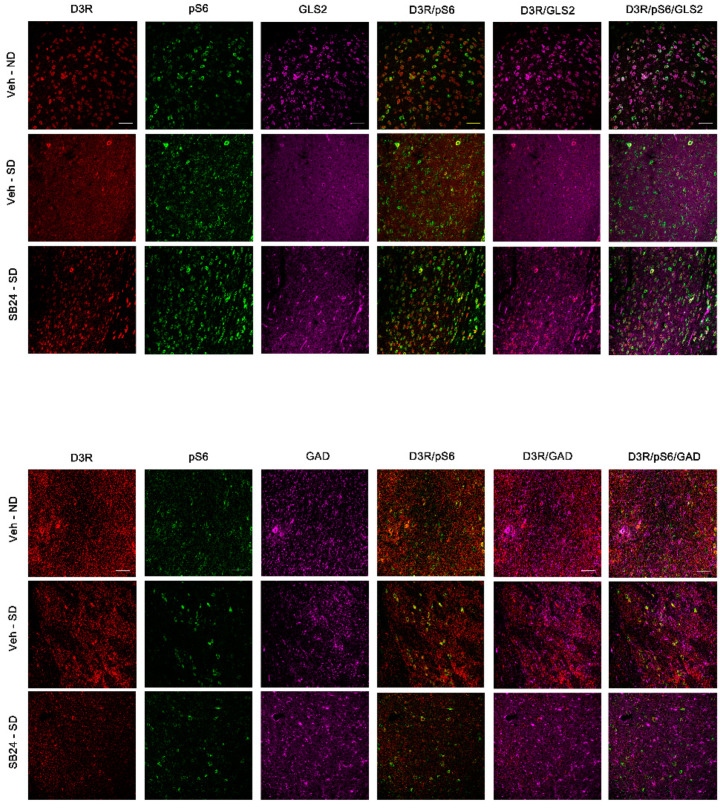
Representative confocal images showing the basolateral amygdala coronal sections immunostained for dopamine type 3 receptor (D3DR, red), phosphorylated S6 protein (activated; green), GLS2 (magenta, glutamatergic neurons), and GAD (magenta, GABAergic neurons) from controls (veh-ND), from mice exposed to a social defeat agonistic encounter after 60 days of abstinence (veh-SD) and from mice receiving an injection of the D3DR antagonist SB-277011-A (24 mg/kg i.p.) prior to the social-defeated agonistic encounter after 60 days of abstinence (SB24-SD). Colocalization of D3DR/GLS2, pS6/GLS2, DRD3/pS6/GLS2, D3DR/GAD, pS6/GAD, and DRD3/pS6/GAD are also shown in the figure. Scale bars: 50 μm.

**Figure 8 ijms-22-03100-f008:**
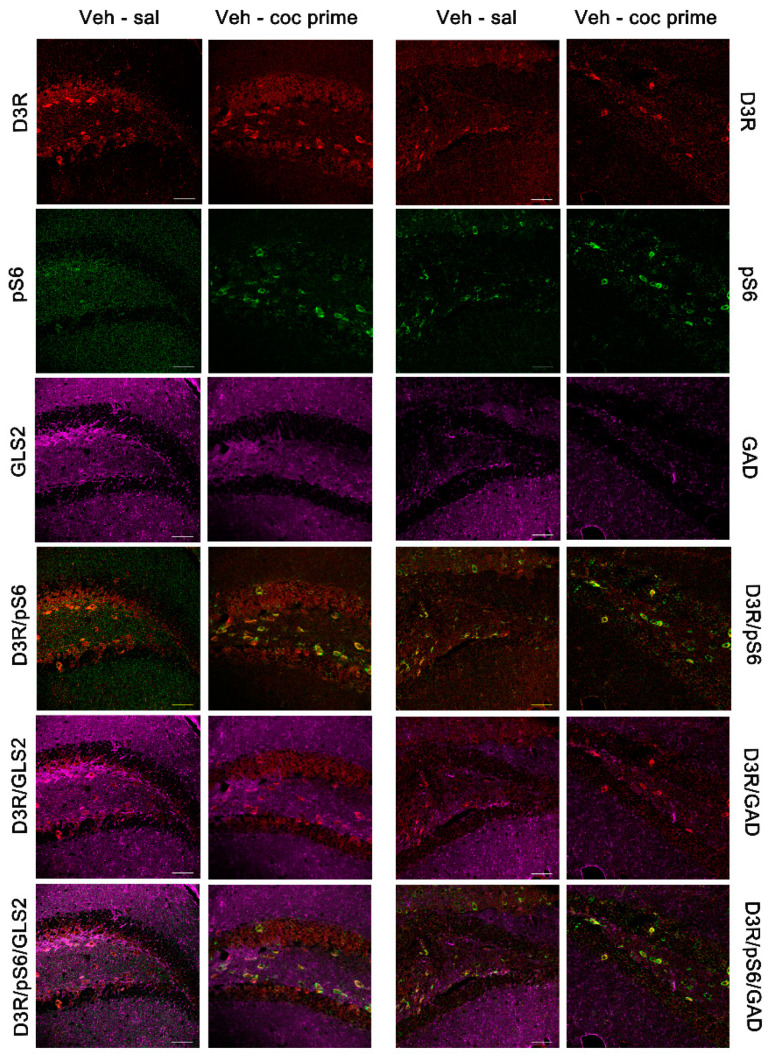
Representative confocal images showing the dentate gyrus coronal sections immunostained for dopamine type 3 receptor (D3DR, red), phosphorylated S6 protein (activated; green), GLS2 (magenta, glutamatergic neurons), and GAD (magenta, GABAergic neurons) from controls (veh-sal) and from mice receiving a cocaine-prime dose after 60 days of abstinence (veh-coc prime). Colocalization of D3DR/GLS2, pS6/GLS2, DRD3/pS6/GLS2, D3DR/GAD, pS6/GAD, and DRD3/pS6/GAD are also shown in the figure. Scale bars: 50 μm.

**Figure 9 ijms-22-03100-f009:**
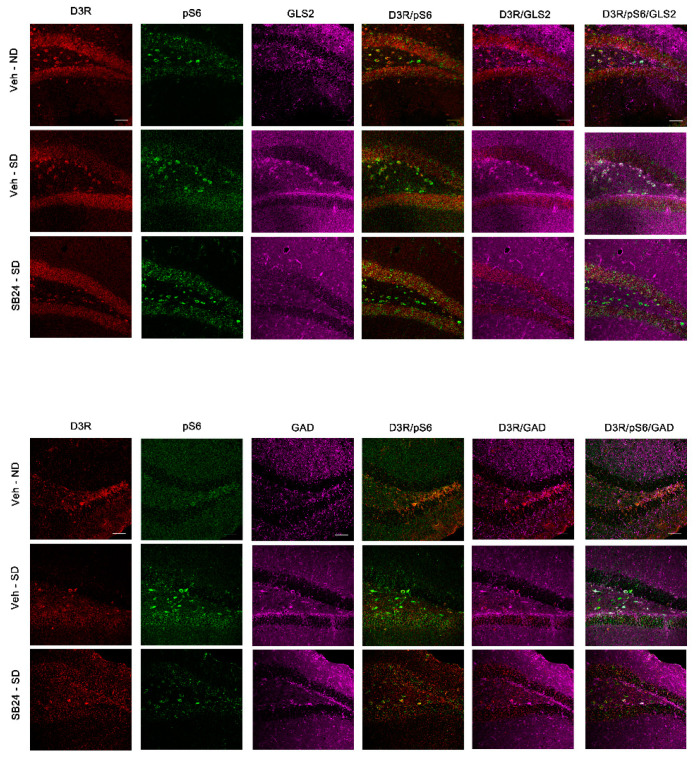
Representative confocal images showing the dentate gyrus coronal sections immunostained for dopamine type 3 receptor (D3DR, red), phosphorylated S6 protein (activated; green), GLS2 (magenta, glutamatergic neurons), and GAD (magenta, GABAergic neurons) from controls (veh-ND), from mice exposed to a social defeat agonistic encounter after 60 days of abstinence (veh-SD) and from mice receiving an injection of the D3DR antagonist SB-277011-A (24 mg/kg i.p.) prior to the social-defeated agonistic encounter after 60 days of abstinence (SB24-SD). Colocalization of D3DR/GLS2, pS6/GLS2, DRD3/pS6/GLS2, D3DR/GAD, pS6/GAD, and DRD3/pS6/GAD are also shown in the figure. Scale bars: 50 μm.

## Data Availability

The data are available from the corresponding authors on reasonable request.

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
