# Peer review of "Distinct Regulation of Dopamine D3 Receptor in the Basolateral Amygdala and Dentate Gyrus during the Reinstatement of Cocaine CPP Induced by Drug Priming and Social Stress"

_ijms, 2021, doi:10.3390/ijms22063100_

Round 1
Reviewer 1 Report
Major comments: 1. The authors used CPP paradigm to test whether SB-277011-A could block the reinstatement induced by acute cocaine challenge or social stress. The experiment is well-designed. However, there are some flaws in the presentation of the results. CPP means Conditioned Place Preference, the animals prefer to spend more time in the chamber where they received addictive drugs. Thus, the ratio of preference (time spent in drug chamber/ total time in both chambers) is more important than the time spent in drug chamber. In the article, the authors only presented the time spent in coc-paired chamber, how about the ratio of preference? It is possible that the time in coc-paired chamber increased but there was no significant difference in the ratio of preference. Furthermore, how about the entry times of each chamber? 2. " Immediately after social defeat, the reinst test was performed.' Could the authors give a specific interval?Author Response
We are grateful for the interesting comments raised by the Reviewers, which helped us to considerably improve the quality of our manuscript. Please find below a point-by-point response to specific comments.
According to the Reviewer comments, we have calculated the ratio of preference for all the sets of animals and have included these data in the Results section and in the Figures 1 and 2, together with the results of the time spent by mice in the cocaine-associated chamber.
As the Reviewer suggested, we have also analyzed the number of entries to the cocaine-associated chamber and also the ratio between the number of entries to the cocaine-associated chamber and the total number of entries to both compartments. However, these analyses have not revealed consistent significant differences between groups nor trends between the pre-conditioning, post-conditioning and extinction tests. Hence, we have not included these data in the manuscript.
Finally, we have specified the interval of seconds (from 5 to 10 s) between the social defeat session and the beginning of the reinstatement test.
Reviewer 2 Report
The authors presented the results of studies on the participation of the dopamine D3 receptor in the cocaine relapse in two modes: by drug priming and social stress (the cue-environment association). The research method used is well chosen, the results are well illustrated and described. The authors investigated behavioural and neurobiochemical changes on the animal model. The reproduced results are consistent, well-described. The data presentation is clear. They can make a significant contribution to understanding cue-paradigm in the cocaine addiction.
To fully appreciate the authors' work, they should complete the missing data:
Minor revision:
The statistics data are incomplete, the value of n is missing in the statistical descriptions of the figures. In the text, it is better to enter the p value as “<”or”>” 0.05 or 0.01 instead of writing the value, for example, p=0.048 (e.g section 3.2). The authors should pay attention to the notation of numbers: often there is a comma instead of a dot in the decimal places.
In the description of the antibodies monoclonal antibodies are marked - does this mean the use of polyclonal antibodies where this information is not available? (section 2.5)
The authors should review the references cited - entries 37 and 39 are the same. It is worth standardizing citations. (Section References).
The work is very interesting, however, the authors must pay special attention to inconsistencies and errors before publication.
Author Response
We are grateful for the interesting comments raised by the Reviewers, which helped us to considerably improve the quality of our manuscript. Please find below a point-by-point response to specific comments.
As reviewer indicates, we have included the sample size in the figure legends. Also, the p value is expressed as > 0.05, < 0.001, < 0.01 or < 0.05, and all the commas in the decimal places have been changed by dots in the description of the results.
Also, according to the reviewer suggestions, we have indicated the origin (monoclonal or polyclonal) of all the antibodies used in the present work.
Lastly, we have eliminated the repeated reference and reviewed the bibliography to find and correct any inconsistence.
Round 2
Reviewer 1 Report
The authors adequately explained my concerns.